# Convolutional Neural Network-Based Clinical Predictors of Oral Dysplasia: Class Activation Map Analysis of Deep Learning Results

**DOI:** 10.3390/cancers13061291

**Published:** 2021-03-14

**Authors:** Seda Camalan, Hanya Mahmood, Hamidullah Binol, Anna Luiza Damaceno Araújo, Alan Roger Santos-Silva, Pablo Agustin Vargas, Marcio Ajudarte Lopes, Syed Ali Khurram, Metin N. Gurcan

**Affiliations:** 1Center for Biomedical Informatics, Wake Forest School of Medicine, Winston-Salem, NC 27101, USA; hbinol@wakehealth.edu (H.B.); mgurcan@wakehealth.edu (M.N.G.); 2School of Clinical Dentistry, The University of Sheffield, Sheffield S10 2TA, UK; h.mahmood@sheffield.ac.uk (H.M.); s.a.khurram@sheffield.ac.uk (S.A.K.); 3Oral Diagnosis Department, Semiology and Oral Pathology Areas, Piracicaba Dental School, University of Campinas (UNICAMP), Bairro Areão, Piracicaba 13414-903, São Paulo, Brazil; a190793@dac.unicamp.br (A.L.D.A.); alan@unicamp.br (A.R.S.-S.); pavargas@fop.unicamp.br (P.A.V.); malopes@fop.unicamp.br (M.A.L.)

**Keywords:** oral epithelial dysplasia, oral cancer, squamous cell carcinoma, leucoplakia, erythroplakia, transfer learning, deep learning, class activation map analysis

## Abstract

**Simple Summary:**

Oral cancer/oral squamous cell carcinoma (OSCC) is among the top ten most common cancers globally; early and accurate diagnosis of oral cancer is critical. Despite improvement in surgical and oncological treatments, patient survival has not improved over the last four decades. Our purpose is to develop a deep learning method to classify images as “suspicious” and “normal” and to highlight the regions of the images most likely to be involved in decision-making by generating automated heat maps. Thus, by using convolutional neural network-based clinical predictors, oral dysplasia in an image can be classified accurately in an early stage.

**Abstract:**

Oral cancer/oral squamous cell carcinoma is among the top ten most common cancers globally, with over 500,000 new cases and 350,000 associated deaths every year worldwide. There is a critical need for objective, novel technologies that facilitate early, accurate diagnosis. For this purpose, we have developed a method to classify images as “suspicious” and “normal” by performing transfer learning on Inception-ResNet-V2 and generated automated heat maps to highlight the region of the images most likely to be involved in decision making. We have tested the developed method’s feasibility on two independent datasets of clinical photographic images of 30 and 24 patients from the UK and Brazil, respectively. Both 10-fold cross-validation and leave-one-patient-out validation methods were performed to test the system, achieving accuracies of 73.6% (±19%) and 90.9% (±12%), F1-scores of 97.9% and 87.2%, and precision values of 95.4% and 99.3% at recall values of 100.0% and 81.1% on these two respective cohorts. This study presents several novel findings and approaches, namely the development and validation of our methods on two datasets collected in different countries showing that using patches instead of the whole lesion image leads to better performance and analyzing which regions of the images are predictive of the classes using class activation map analysis.

## 1. Introduction

The lack of an objective clinical method to evaluate oral lesions is a critical barrier to the early, accurate diagnosis and appropriate medical and surgical management of oral cancers and related diseases. This clinical problem is highly significant because oral cancer has an abysmal prognosis. The 5-year survival for Stage IV oral squamous cell carcinoma (OSCC) is only 20–30%, compared to 80% for Stage I (early) OSCC. A significant proportion (more than 50%) of OSCCs are preceded by oral epithelial dysplasia (OED) or oral pre-cancer. Therefore, early detection of pre-cancerous oral lesions (with histological evidence of dysplasia), and reversal of habits such as smoking, tobacco chewing, reducing alcohol consumption, and surgical management can significantly reduce the risk of transformation into cancer [1,2]. Thus, early detection is critical for improving survival outcomes. 

There are no individual or specific clinical features that can precisely predict the prognosis of an oral potential malignant lesion. However, an improved prediction may be accomplished by jointly modeling multiple reported “high-risk features”, such as intraoral site (i.e., lateral tongue borders, ventral tongue, the floor of the mouth, and lower gums are high-risk sites), color (red > mixed > white), size (>200 mm^2^), gender (females > males), appearance (erythroplakia and verrucous lesions represent high risk), and habits (tobacco and alcohol usage) [1,2,3]. At present, clinical assessment of these lesions is highly subjective, resulting in significant inter-clinician variability and difficulty in prognosis prediction, leading to suboptimal quality of care. Thus, there is a critical need for novel objective approaches that can facilitate early and accurate diagnosis and improve patient prognosis. 

Diagnosis of clinically suspicious lesions is confirmed with a surgical biopsy and histopathological assessment [4,5]. However, the decision to refer a patient to secondary care or perform a biopsy depends on the practitioner’s clinical judgment. It is based on subjective findings from conventional clinical oral examination (COE). Unfortunately, COE is not a strong predictor of OSCC and OED, with a 93% sensitivity and 31% specificity, highlighting the need for objective and quantitative validated diagnostic methods [4,5,6]. 

Recently, convolutional neural network (CNN)-based image analysis techniques have been used to automatically segment and classify histological images. Santos et al. presented a method for automated nuclei segmentation on dysplastic oral tissues from histological images using CNN [7] with 86% sensitivity and 89% specificity. Another CNN-based study proposed a framework for the classification of dysplastic tissue images to four different classes with 91.65% training and 89.3% testing accuracy using transfer learning [8,9]. Yet another CNN-based transfer learning approach study proposed by Das et al. [10] also classified the multi-class grading for diagnosing patients with OSCC. They used four of the existing CNN models, namely Alexnet [11], Resnet-50 [12], VGG 16, and VGG 19 [13], to compare their proposed CNN method, which outperformed all the models with 97.5% accuracy. Although these studies show that histological images can be classified accurately, predicting the lesion’s risk of malignant progression on clinical presentation or images is crucial for early detection and effective management of lesions to improve the survival rates and prevent oral cancer progression [4]. 

Radiological imaging modalities such as magnetic resonance imaging (MRI) and computed tomography (CT) can help determine the size and extent of an OSCC prior to surgical intervention. However, these techniques are not sensitive enough to detect precancerous lesions. To overcome this barrier, a range of adjuvant clinical imaging techniques have been utilized to aid diagnosis, such as hyperspectral imaging (HSI) and optical fluorescence imaging (OFI), and these images have the potential to be analyzed using computer algorithms. Xu et al. presented a CNN-based CT image processing algorithm to classify oral tumors using 3DCNN instead of 2DCNN [14]. The 3DCNN method had better performance than 2DCNN, because the spatial features of the three-dimensional structure extract tumor features from multiple angles. Jeyaraj and Nadar proposed a regression-based deep CNN algorithm for an automated oral cancer-detecting system by examining hyperspectral images [15]. Comparison of the designed deep CNN performance was better than other conventional methods such as support vector machine (SVM) [16] and deep belief network (DBN) [17]. These methods segment intraoral images accurately and classify the inflamed gingival and healthy gingival automatically. However, they require HSI and OFI, which are not commonly available in dental screening. 

Some studies have explored both autofluorescence and white light images captured with smartphones. Song et al. presented a method for the classification of dual-modal images for oral cancer detection and used a CNN-based transfer learning algorithm with 86.9% accuracy. However, the ground truth of the diagnosis depends on the specialist results rather than the histopathological results [18]. Uthoff et al. proposed a system to classify “suspicious” lesions using CNN with 81.2% accuracy, but the system needs an additional Android application, an external light-emitting diode (LED) illumination device [19,20]. Using fluorescence imaging, Rana et al. reported pixel-wise segmentation of the oral lesions with the help of CNN-based autoencoders [21]. It was proposed in a review that OFI is an efficient tool for COE in managing oral potentially malignant disorders (OPMD) [22]. The review provided contemporary evidence in support of using OFI during COE for diagnosis and prognosis purposes. However, these studies require autofluorescence or hyperspectral imaging. These modalities are not widely available and are difficult to interpret, therefore limiting their use in early detection of oral cancer or dysplasia.

One of the more recent studies focused on lesion detection and a classification system using white-light images obtained from mobile devices [23]. This system, which used ResNet-101 [12] for classification and Fast R-CNN for object detection, achieved an F1-score of 87.1%. While the performance is encouraging, the results are not interpretable. The method in [23] requires both object detection and segmentation; however, their object detection F1-score is only 41.18% for the detection of lesions that required referral. 

In our application, photographic images of oral lesions were manually annotated as “suspicious” and “normal” areas to develop an automated classification methodology (see Section 2.1). We implemented a CNN-based transfer learning approach, using the Inception-ResNet-v2 pre-trained network and compared the results with those obtained with VGG-16 and Resnet-101 pre-trained networks (see Section 3. In our study, we used only photographic images instead of fluorescence or hyperspectral images. 

We also analyzed which regions of the images were used to predict the classes. This analysis is important in understanding how the neural networks analyze images, hence gaining insight into why the neural network misclassifies certain images (see Section 3). Finally, we compared the system’s performance when trained with image tiles versus regions of interest. We validated our system by 10-fold cross-validation and leave-one-patient-out validation techniques and presented the performance results in terms of accuracy, F1-score, recall, and precision (see Section 3).

## 2. Materials and Methods

In this study, we developed a system for the classification of potentially malignant oral lesions into normal and suspicious categories from photographic images of oral lesions and their surrounding normal areas. The inputs to the system, the block diagram of which is shown in Figure 1, are photographic images, and the outputs are heat maps of the classified images with their classification categories.

### 2.1. Dataset

For this study, two datasets were used: the Sheffield (UK) dataset and Piracicaba (Brazil) dataset. These images were obtained after appropriate ethical approvals. Ethical approval for the Sheffield cohort was obtained for this study from the HRA and Health and Care Research Wales (HCRW), Reference number 18/WM/0335 on 19 October 2018. This study was approved by the Piracicaba Dental Ethical Committee, Registration number 42235421.9.0000.5418, on 9 June 2019. Material transfer agreements were approved and put in place between the three research sites for sharing of anonymized clinical images. The images were standard color images captured with a regular photographic cameras (see Appendix A) capturing the lesion as well as the surrounding regions of the mouth. Depending on where the lesion was located, the surrounding area could include teeth, dental mirror/retractor, and some parts of the face (e.g., mouth or chin). The Sheffield dataset included 30 images with a known diagnosis for each of the three grades of dysplasia: mild (10 images), moderate (10 images), and severe (10 images). In the photographs, each lesion was visually identified by the clinical team members, and precise boundaries were drawn using Aperio/Leica ImageScope version 12.4.0 software byLeica Biosystems Imaging, Inc. from Buffalo Grove, IL, USA. Additionally, normal-appearing mucosal areas near the lesion were also annotated using a different color to provide a frame of reference for algorithm development (see Figure 2). This “normal” annotation is vital because each patient’s mouth lining/mucosa color may differ. While a visual assessment can indicate the possibility of dysplasia, the specific dysplasia grade (i.e., mild, moderate, or severe) can only be decided after a biopsy and histopathological exam. The Piracicaba dataset, collected from 24 patients, contains 43 images with a known diagnosis for each of the three grades of dysplasia, namely mild (11 images), moderate (9 images), and severe (4 images) and is annotated in the same manner. For each patient (54 patients in total in both datasets), the identified lesions and normal mucosal areas (RoI: Region of Interest) are presented in Table 1.

We used two different cross-validation methods. For the first method, the datasets were divided into 10-fold (k = 10 folds cross-validation) randomly, but considering that all images from a patient were used in either test or training/validation sets. For each fold, 10% of the patient’s images were randomly selected as test images. The rest of the data were divided into training and validation sets with a ratio of 85% and 15%, respectively. The second cross-validation method was leave-one-patient-out (LoPo). While testing the system, only one of the patient’s images was tested, and the rest of the dataset was used for training (85%) and validation (15%). 

To test the system, we used two approaches: test the patch and test the patient. “Test the patch” means that each patch in the fold is classified. The system’s accuracy is calculated depending on the number of correct classified patches (normal or suspicious). “Test the patient” means that the accuracy is calculated for the RoI instead of each patch. If the number of correct classified patches for each RoI is greater than the number of wrong classified patches, then that RoI is accepted as being correctly classified; otherwise, it is considered a misclassification. Since each RoI is divided into patches, it is more meaningful to consider the combination of these patches (rather than single patches) to decide on the classification result of the RoI.

### 2.2. Preprocessing and Data Augmentation

The system has a preprocessing step to prepare images before classifying them as “normal” or “suspicious” (see Figure 3). First, we extracted small patches of size 128 × 128 pixels (see Figure 4). If the patches with more than 80% pixels originated from a lesion area, they were labeled as suspicious patches; otherwise, they were considered normal patches. 

Because the number of cases was limited, we applied data augmentation for the images. Both for suspicious and normal patches, data augmentation was performed with random horizontal and vertical flips, randomly shifting an input patch in the horizontal and vertical directions with two offsets sampled uniformly between −50 and 50 pixels, rotating the images starting from −5 to 5 and increasing by 5 up to −45 to 45 angles, sharpening images (alpha ranged between 0 to 1 and lightness between 0.75 and 1.5), randomly scaling the inputs by a factor between 0.5 and 1.5, and elastic transformation (alpha ranged between 45 and 100, and sigma was 5) [24]. (We used the predefined “imgaug” library for image augmentation in Python.) Figure 5 gives examples of augmented image patches. We obtained 6100 lesion and 6000 normal mucosa patches for the Sheffield dataset and 7314 suspicious and 7370 normal mucosa patches for the Piracicaba dataset after augmentation.

### 2.3. Classification Approach

To classify patches as normal or suspicious, we used deep CNN, which has been successfully applied in image classification, segmentation, object detection, video processing, natural language processing, and speech recognition [25,26]. For oral cancer prognosis or diagnosis, CNN has been applied to histological images [9,27,28,29] and on autofluorescence and white light images [18,19]. To our knowledge, this is the first application of CNN for the classification of oral lesions from clinical photographic images. Additionally, we, for the first time, interpreted class activation maps resulting from different pre-trained convolutional neural networks: Inception-ResNet-v2, Inception v3, ResNet101, and VGG16.

### 2.4. Transfer Learning

To train a deep CNN, a huge number of images are required. Because of the limited number of image samples for oral dysplasia, we used transfer learning, which uses features learned from a problem in one domain and applies them to a problem in a different domain [30,31,32,33]. For example, in Aubreville et al.’s works [27,29], Inception V3 [34], previously trained on the ImageNet dataset [11], was retrained to detect one of the cancer types of the epithelium automatically. Song et al.’s work [18] also presented an image classification approach based on autofluorescence and white light images using transfer learning with networks VGG-CNN-M [35]. Welikala et al. [23] used deep learning-based photographic image classification of oral lesions for early detection of oral cancer using ResNet-101 pre-trained network. However, none of these applications made an effort to develop class activation maps to understand how the network makes a decision. To compare the results, we also tested the performance of the previous studies’ pre-trained networks with the same structure (i.e., the number of the frozen and fine-tuned layers was kept the same) on our dataset images.

In addition to testing the previous studies’ pre-trained networks, we implemented a transfer learning approach by retraining the last seven layers of a pre-trained Inception-ResNet-v2 network [36]. This network was trained and validated with 50,000 images to classify 1000 object categories, learning-rich feature representations, with 825 layers. We did not retrain the whole network because it was highly likely to result in overfitting [37,38]. Instead, we opted to freeze the first 818 layers, a number that was decided empirically to limit the number of parameters required to learn the features in the network. Other pre-trained networks’ retraining structures were regenerated with the first 307 frozen layers of Inception-v3 and 79 frozen layers of ResNet-101; however, VGG-16 was re-trained without any frozen layers. The number of frozen layers and the number of parameters that were retrained are stated in Table 2. We retrained the last three layers (prediction, softmax, and classification) of the pre-trained network with patches of oral images in our database. Figure 1 shows the block diagram of transfer learning layers frozen, fine-tuning, and the changes we made to the network to solve our problem.

### 2.5. Class Activation Maps (CAM)

To analyze which parts of an image a neural network focuses on while making a classification decision, class activation maps (CAMs) are used for various applications, including cancer classification [39], grading [40], and diagnosis [41]. CAM is generated for each class of the network by obtaining the weighted sum of the last convolutional features (activation maps) using the fully connected layer weights [42]. For each category to be classified by the CNN, the activation maps indicate which parts of the image are effective in classification. In our case, after the network was retrained and the weights of the features were updated, a heat map of the predicted class was generated by creating the class activation map. We created only the predicted class’s heat map to understand which spatial regions are effective in making a right or wrong decision.

### 2.6. Experimental Setup

To classify the oral images as normal or suspicious, we used the following training parameters. The numbers of the frozen layers are mentioned for each pre-trained CNN in Section 2.4. The cross-entropy cost function was optimized with stochastic gradient descent [43], with a batch size of 64 samples taken from training images. The learning rate was initially assigned as 3 × 10^−4^. The number of epochs was a maximum of 20, but to avoid overfitting, the system stopped training if there was no improvement for more than ten iterations. The implementation was done in MATLAB 2019b (Mathworks, Natick, MA, USA) using the Deep Learning Toolbox. 

The steps that affect the experimental results are explained in detail in Section 2.1 and Section 2.2: augmentation and cross-validation [44]. Data augmentation is a pre-processing step performed for both system training and testing; therefore, patch images were augmented. The training and testing processes depend on the cross-validation for either ten-fold or LoPo cross-validation methods to split the data to perform the experiments.

## 3. Results

We compared the pre-trained CNN results, namely Inception ResNetV2, InceptionV3, VGG16, and ResNet-101, for both accuracy and F1-score after retraining them on two different datasets. Ten-fold cross-validation and leave-one-patient-out cross-validation results for the Sheffield and Piracicaba datasets are stated in Table 3 and Table 4, respectively.

As seen in Table 3 and Table 4, the training and validation accuracies were similar but less variable (i.e., smaller standard deviation) than those in the test patch set. For example, the Inception-ResNet-V2 showed average accuracies of 81.1% and 80.9% for the Sheffield dataset training and validation, respectively, whereas the test accuracy was 71.6%. The patient-level average and maximum test accuracies were higher than the patch-level accuracies. For example, for the same cases, the patch-level accuracy was 71.6%, whereas the patient-level accuracy was 73.6%.

The minimum values of ten-fold test accuracies were higher than those of the LoPo test accuracies. The ten-fold average test results for the Piracicaba dataset were higher than those of the Sheffield dataset and closer to the validation and training accuracies. These results mean that the classifiers trained on the Sheffield dataset were prone to overfitting. 

The highest accuracy was achieved among the different networks by ResNet-101 for ten-fold, and LoPo cross-validation approaches. VGG16 performance for LoPo cross-validation of the Piracicaba dataset showed the highest average accuracy; however, the standard deviation was higher than that of the ResNet-101, and the standard deviation and the accuracy values with patches were higher than those with the VGG-16 patches.

The F1-score, recall, and precision for patches, patients, and RoI are presented in Table 5. The trends are similar to those shown in Table 4 because ResNet-101 and VGG-16 F1-scores for the Piracicaba dataset are among the highest. While ResNet-101 results in the highest RoI values, VGG-16 has the highest patient values. Because the number of regions can vary from patient to patient, ResNet-101 results are more accurate than all the others. 

After ten-fold and LoPo cross-validation tests were performed for each dataset, the Piracicaba dataset was used as a test set. The Sheffield dataset was used for training and validation, and vice-versa. The patch- and patient-based test results are shown in Table 6. Again, the Piracicaba dataset results appear more accurate than that of the Sheffield dataset. This difference in accuracy may be explained by many factors, including the differences in the number of images, RoIs, image size and quality, and where the patches are selected. The image dimensions and bit depths are smaller, affecting the sizes of suspicious lesions.

To understand which regions of the image affect the classification result, we represented the regions with a heat map using the CAM method, as explained in the methods section. To show the images’ heat map, we performed the classification on RoI as a single image, and the results of the heat maps of the classification are shown in Figure 6. As seen in the figure, for cases that were classified as “suspicious", the white and partially white regions were more effective in classification, with scores of 0.68 and 0.63, respectively. For the normal cases, shown in the figure, the upper image has white light reflections. These regions are mostly colored with black, blue, and green in the heat map for “normal” classification, indicating less suspicion. The other regions where the heat map is colored with yellow to red are the regions that result in normal classification. These examples demonstrate the system’s success in classifying “suspicious” and “normal” oral images using the features extracted from the associated regions.

While the heat map correctly identified all the correct areas, as in Figure 6 for both normal and suspicious cases, in some samples the results of the heat map were somewhat misleading, for example in Figure 7. In this case, the heat map did not mark all the regions of interest correctly. However; if we divide the image into patches, the results are more meaningful, as shown in Figure 7. Therefore, our approach was to divide the images into patches to improve their classification and also to be able to interpret the results of deep learning better.

In Figure 8, the upper suspicious image shows some part of a tooth, the color of which caused the classifier to misclassify the image. CAM analysis showed that the lesion’s most worrying clinical part was not colored as highly suspicious (red) but moderately suspicious (yellow). For a less suspicious image, the red and yellow parts of the heat map did not represent the lesion, and the lesion was not colored as yellow or red.

We also investigated the heat maps of the misclassified images to understand which regions affect the classification of the oral images. Two of the misclassified images are shown with their heat map representations in Figure 8. Both of the images were classified as suspicious, whereas they were normal. Their suspicion scores were 0.57 and 0.55, and the red regions were not seen as suspicious or different to normal regions. These borderline incorrect cases could potentially be explained by the small datasets that were used to train the classifier.

The differences in the heat map of ResNet-101 and Inception-Resnet-v2 were investigated for two specific images. The images were predicted accurately, but the lesion region area on the images was not marked as suspicious on the image (see Figure 8). As seen in Figure 9, ResNet-101 heat maps were more meaningful than those of the Inception ResNet-v2 results in showing the more suspicious parts of the lesion. However, some parts were not marked as highly suspicious in the heat map but more so than the other regions of the image. For instance, as seen in the image on the right side in Figure 9, ResNet-101 classified the image as suspicious with a score of 0.99; however, the yellow region of the upper left side may be marked with the orange to red scale on the heat map.

## 4. Discussion

This study shows that the maximum accuracy of the classification of oral images was 95.2% (± 5.5%) for 10-fold and 97.5% (±7.9%) for LoPo cross-validation approaches with the ResNet-101 pre-trained network and the Piracicaba dataset. Additionally, the maximum accuracy was 86.5% on an independent dataset (Piracicaba) for patient-based test results with Inception ResNet-v2 when the Sheffield dataset was used for training. When the Piracicaba dataset was used for training and the Sheffield dataset was used for testing, ResNet-101 network results were more accurate than those of Inception ResNet-v2. This shows that for different datasets, ResNet-101 is not accurate in each test set, and the results with this deep learning method do not generalize to all datasets. However, for both of the results performed on these pre-trained networks, the system was more accurate when the Sheffield dataset was used for training and the Piracicaba dataset was used for testing. An explanation for this is that when the Sheffield dataset is used for training, the system is trained on relatively lower quality and more challenging images, and the resulting classifier works well on the higher-quality Piracicaba images. However, when the system is trained with better quality images, its performance is lower for the relatively lower quality images. 

We used precision, recall, and F1-score to measure the generalizability of the system (see Table 5). Presenting the results in accuracy, F1-score, precision, and recall evaluation methods allowed us to compare our results with other studies. The F1-score ranged from 69.8% to 97.9%; recall and precision varied between 67.3% and 100%, and 78.2% and 100%, respectively. The highest F1-score was obtained for the Piracicaba dataset, with VGG-16 pre-trained models for overall performance and the other three pre-trained networks performing similarly well. For the RoI-level results, the best F1-score was obtained with ResNet-101.

Studies similar to ours in the literature have used both autofluorescence and white light images together [18,19], but the results show that for white light images (which are close to our photographic images), the performance was the least accurate. In these dual-modality methods, the most accurate result was 86.9%. It is hard to compare this result to ours because they used more than a single image source and did not report their independent test set results. The dual-modality system’s sensitivities (recall) and positive predictive values (precision) were 85.0% and 97.7%, respectively [19] and all predictive values ranged from 67.3% to 100%. Because of the limited diversity of the datasets and the small number of cases, our reported standard deviation values were higher than those reported by Uthoff et al.’s study [19], which also used multi-modality (but not clinically relevant or easily available) images to train and test their performance.

Our study does have some limitations. All of our results were derived from 54 patients in total, which is a small number to train and test the system independently; however, it was sufficient to demonstrate the feasibility of our approaches. In order to overcome this limitation, we increased the number of images by augmentation and split the images into patches. This study also demonstrated that the classification accuracy could be increased by extracting patches from the oral images. 

Another limitation is in the selection of the patches, during which we used a percentage threshold for the number of the total pixels to decide whether a patch is suspicious or normal. Some studies segment lesions automatically, but we used manually segmented regions. Manual segmentation could be prone to inter- and intra-reader variability. In our future studies, we aim to overcome this shortcoming by developing automated segmentation methodologies. 

With an independent dataset accuracy of 86.5% and an F1-score of 97.9%, the results are promising, especially considering that the networks were developed and tested using small datasets. However, to have better results and develop a more generalizable method, the dataset’s size needs to be increased. With a bigger cohort, we aim to subclassify suspicious images as “high risk” or “low risk.”

After developing automated segmentation and sub-category prediction algorithms using photographic images, we are planning to combine these results with the analysis of histology images. Additionally, we will further enhance our system by including age, gender, lesion location, and patient habits as potential risk factors as well as analyze the follow-up information to predict the risk of malignant transformation. We expect that combining these clinical features with the analysis results from photographic and histological images will result in a very comprehensive clinical decision support tool.

## 5. Conclusions

In this study, we proposed a deep CNN-based oral lesion classification system for clinical oral photographic images. To retrain the pre-trained models adapted to our system, we divided the annotated oral images into square regions (patches) and analyzed which regions are used in the prediction of the classes. The proposed system is one of the rare studies that uses only photographic images and is the only study that shows the heat maps of the images according to the class activation maps. 

This study is also the first to use datasets collected at two different institutions in different countries to analyze the variation. One dataset was used for training and the other one was used for testing the performance changes. When independently tested, the Piracicaba dataset results appeared to be more accurate than those of the Sheffield dataset in 10-fold and LoPo cross-validation approaches. While this may be due to the relatively small datasets, it also highlights the importance of testing these algorithms on different datasets to measure their generalizability.

## Figures and Tables

**Figure 1 cancers-13-01291-f001:**
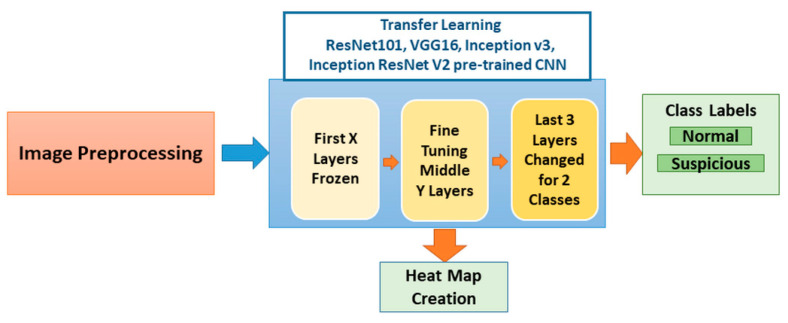
Block diagram of the system. The image preprocessing stage starts with patching the image and continues with the augmentation of the image patches. Pre-trained Inception-ResNet-v2 was retrained with these images. Classification and heat map generation were performed. Four pre-trained networks were independently used to analyze the results. Each network has a different number of layers (ResNet101→347-layer, VGG16→41, Inception v3→315-layer, and Inception ResNet v2→824-layer) with variation in frozen layers (ResNet101→79-layer, VGG16→no frozen, Inception v3→307-layer, and Inception ResNet v2→818-layer) and fine-tuning of middle layer numbers; we represent them as X and Y in the figure). CNN: convolutional neural network.

**Figure 2 cancers-13-01291-f002:**
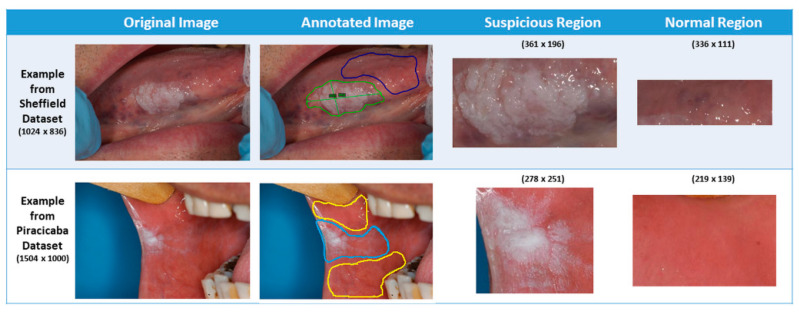
Differentiated “normal” and “suspicious” bounding box (rectangle) region from the original image according to the lined region.

**Figure 3 cancers-13-01291-f003:**
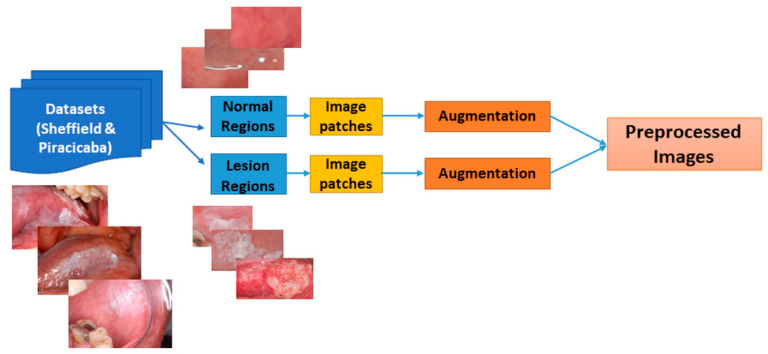
Block diagram of image preprocessing. Two dataset annotated images are cropped from the bounding box of the suspicious and normal regions. Then each image is divided into 128 × 128 image patches. Lastly, the patched images are augmented.

**Figure 4 cancers-13-01291-f004:**
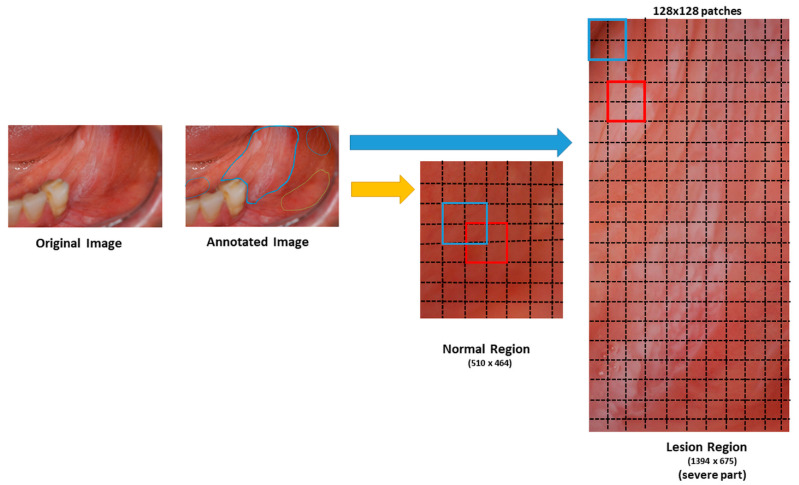
Figurative image patching from bounding box lesion and normal area into 128 × 128 patches. Original images were annotated by dentists. We cropped the bounding box of the annotated regions as normal or suspicious regions. The cropped regions were divided into 128 × 128 pixel image patches (red and blue squares) with a stride of 64 pixels (dotted lines), resulting in overlapping patches. If the size of the image was not a multiple of 128 × 128, the patches were not obtained from the remainder of the image. If the patches with more than 80% pixels originated from a suspicious area, they were labeled as suspicious patches; otherwise, they were considered as normal patches.

**Figure 5 cancers-13-01291-f005:**
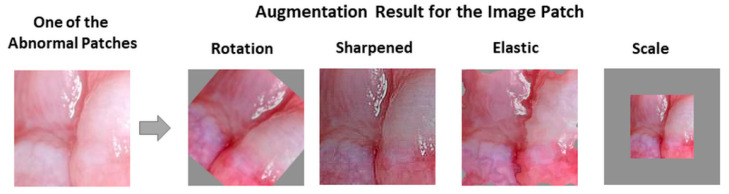
Examples of patch augmentation after rotation, sharpened, elastic, and scale.

**Figure 6 cancers-13-01291-f006:**
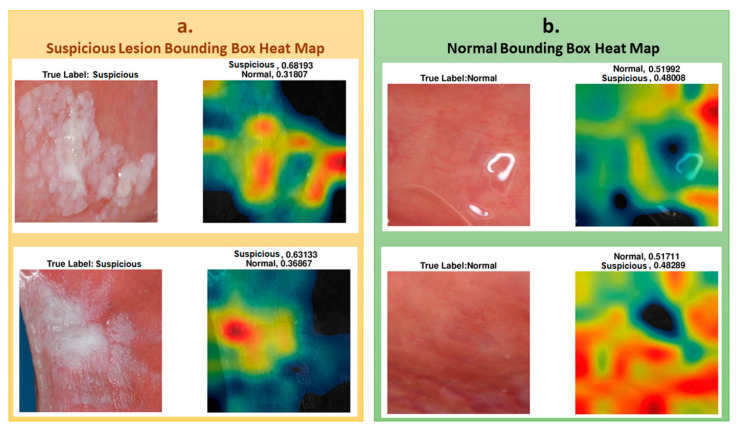
(**a**) Heat maps overlaps the lesion area and accurately classified for suspicious. (**b**) Heat maps overlaps the normal area and accurately classified for normal cases.

**Figure 7 cancers-13-01291-f007:**
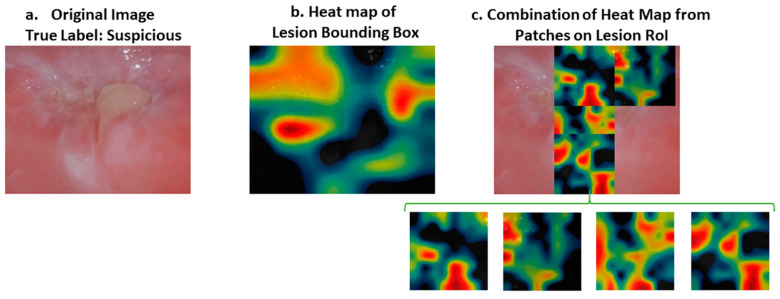
(**a**) Original image. (**b**) Heat map of lesion bounding box. (**c**) Heat map of a combination of patches belonging to the severe part of the lesion area. Original image was classified as “suspicious” when the region of interest was tested. However, the heat map of the region of interest did not reflect the severe part of the image with a warm color (e.g., red). On the other hand, when we divide the severe part of the image into four patches, the heat map of the patches indicates the severe region with an intense color between yellow and red. RoI: Region of Interest.

**Figure 8 cancers-13-01291-f008:**
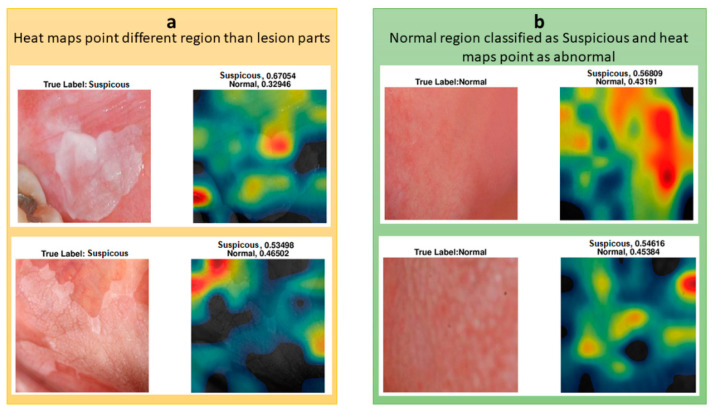
(**a**) Heat maps do not point to lesion region. (**b**) Heat maps of misclassified “normal” images.

**Figure 9 cancers-13-01291-f009:**
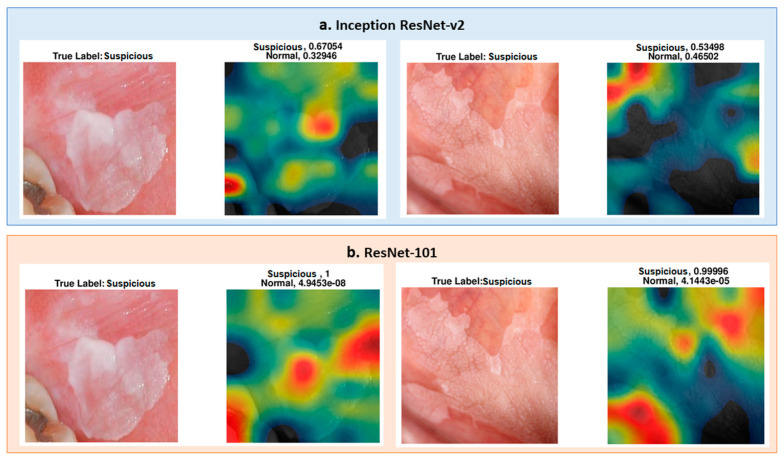
(**a**) Inception ResNet-v2 pre-trained network results for two different “suspicious” cases, which are not well defined. (**b**) ResNet-101 pre-trained network results for the same images of the upper row, and the ResNet-101 heat map is better than the upper row results. However, there are still some parts that are unexpected on the heat map.

**Table 1 cancers-13-01291-t001:** Number of cases and RoI (region of interest) for normal and suspicious.

	Sheffield Dataset		Piracicaba Dataset	
	Number of Cases	Number of RoI	Number of Cases	Number of RoI
Normal	30	68	24	38
Suspicious	30	76	24	57
Total	30	144	24	95
Whole Image Size	1024 × 685		1504 × 1000	

**Table 2 cancers-13-01291-t002:** Number of retrained cross-validation results for Sheffield and Piracicaba datasets.

Pre-Trained Network	Total Number		Number of		
	Layers	Parameters (Millions)	Frozen Layers	Retrained Parameters (Millions)	Image Size
Inception ResNetv2	825	55.9	818	1.5	299 × 299 × 3
Inception-v3	315	23.9	307	2.05	299 × 299 × 3
ResNet-101	347	44.6	79	43.2	224 × 224 × 3
VGG-16	41	138	0	138	224 × 224 × 3

**Table 3 cancers-13-01291-t003:** Ten-fold cross-validation results for Sheffield and Piracicaba datasets.

			Minimum	Maximum	Average	Standard Deviation
Inception ResNet-v2	Sheffield Dataset	Train	76.1%	85.0%	81.1%	2.7%
		Validation	76.7%	85.2%	80.9%	3.0%
		Test Patch	54.9%	88.0%	71.6%	10.0%
		Test Patient	33.3%	95.8%	73.6%	19.0%
	Piracicaba Dataset	Train	80.6%	85.1%	83.1%	1.2%
		Validation	80.8%	84.5%	82.7%	1.1%
		Test Patch	70.1%	92.5%	80.0%	7.5%
		Test Patient	73.7%	100.0%	90.9%	12.0%
ResNet-101	Sheffield Dataset	Train	98.6%	99.4%	99.0%	0.3%
		Validation	97.3%	99.2%	98.7%	0.6%
		Test Patch	74.3%	94.2%	83.6%	7.1%
		Test Patient	83.3%	100.0%	93.2%	7.5%
	Piracicaba Dataset	Train	92.9%	95.8%	94.7%	1.0%
		Validation	92.8%	96.3%	94.4%	1.3%
		Test Patch	60.1%	99.7%	83.9%	13.9%
		Test Patient	87.5%	100.0%	95.2%	5.5%
Inception-v3	Sheffield Dataset	Train	68.7%	83.2%	78.7%	3.7%
		Validation	69.9%	82.9%	78.8%	3.5%
		Test Patch	56.1%	88.8%	71.3%	10.5%
		Test Patient	58.8%	100.0%	83.1%	13.3%
	Piracicaba Dataset	Train	75.2%	80.7%	78.9%	1.8%
		Validation	75.4%	80.3%	78.7%	1.7%
		Test Patch	43.8%	87.8%	72.5%	12.6%
		Test Patient	50.0%	100.0%	81.9%	17.3%
VGG-16	Sheffield Dataset	Train	73.4%	81.7%	78.3%	3.1%
		Validation	73.8%	81.4%	77.7%	3.0%
		Test Patch	47.3%	96.6%	70.8%	13.7%
		Test Patient	81.8%	100.0%	91.2%	8.0%
	Piracicaba Dataset	Train	87.7%	92.9%	91.2%	1.9%
		Validation	87.7%	93.3%	91.0%	1.9%
		Test Patch	74.0%	95.8%	85.4%	8.4%
		Test Patient	72.7%	100.0%	94.0%	9.6%

**Table 4 cancers-13-01291-t004:** Leave-one-patient-out cross-validation results for Sheffield and Piracicaba datasets.

			Minimum	Maximum	Average	Standard Deviation
Inception ResNet-v2	Sheffield Dataset	Train	76.2%	83.2%	80.7%	1.8%
		Validation	76.6%	83.7%	80.6%	1.8%
		Test Patch	15.4%	100.0%	76.7%	16.3%
		Test Patient	50.0%	100.0%	82.1%	21.3%
	Piracicaba Dataset	Train	76.1%	83.0%	81.2%	1.7%
		Validation	77.3%	84.0%	81.4%	1.8%
		Test Patch	45.0%	98.8%	76.0%	14.9%
		Test Patient	0.0%	100.0%	88.4%	23.7%
ResNet-101	Sheffield Dataset	Train	73.5%	83.1%	78.1%	2.5%
		Validation	72.0%	84.6%	77.6%	2.6%
		Test Patch	45.7%	95.6%	72.3%	15.7%
		Test Patient	33.3%	100.0%	75.0%	25.1%
	Piracicaba Dataset	Train	91.9%	94.2%	93.2%	0.7%
		Validation	90.8%	94.3%	92.9%	0.9%
		Test Patch	30.0%	100.0%	87.5%	15.2%
		Test Patient	0.0%	100.0%	97.5%	7.9%
Inception-v3	Sheffield Dataset	Train	70.2%	83.8%	78.5%	2.2%
		Validation	68.3%	84.5%	78.6%	2.6%
		Test Patch	33.3%	95.5%	69.2%	16.9%
		Test Patient	33.3%	100.0%	74.9%	25.3%
	Piracicaba Dataset	Train	78.6%	82.8%	80.6%	1.0%
		Validation	78.2%	83.0%	80.6%	1.2%
		Test Patch	45.8%	98.3%	75.0%	13.9%
		Test Patient	0.0%	100.0%	89.6%	21.4%
VGG-16	Sheffield Dataset	Train	96.4%	99.4%	98.1%	0.7%
		Validation	96.1%	99.6%	97.8%	0.8%
		Test Patch	51.4%	100.0%	78.8%	15.4%
		Test Patient	50.0%	100.0%	85.5%	19.0%
	Piracicaba Dataset	Train	88.0%	95.6%	93.1%	1.8%
		Validation	88.4%	95.6%	92.9%	1.8%
		Test Patch	8.1%	100.0%	84.1%	19.9%
		Test Patient	50.0%	100.0%	98.1%	9.8%

**Table 5 cancers-13-01291-t005:** Leave-one-patient-out cross-validation F1-score, recall, and precision for Sheffield and Piracicaba datasets.

			F1-score	Recall (Sensitivity)	Precision (PPV)
Inception ResNet-v2	Sheffield Dataset	Patches	77.9%	74.1%	90.2%
		Patient	87.2%	99.3%	81.1%
		RoI	87.0%	77.8%	97.6%
	Piracicaba Dataset	Patches	69.8%	66.9%	85.8%
		Patient	94.2%	97.9%	92.5%
		RoI	89.4%	82.1%	98.2%
ResNet-101	Sheffield Dataset	Patches	71.1%	72.0%	79.0%
		Patient	84.0%	98.0%	78.2%
		RoI	80.2%	73.0%	87.0%
	Piracicaba Dataset	Patches	85.0%	67.3%	89.1%
		Patient	94.5%	98.5%	94.9%
		RoI	96.3%	84.8%	98.4%
Inception-v3	Sheffield Dataset	Patches	69.9%	71.3%	78.4%
		Patient	85.1%	95.8%	80.8%
		RoI	78.3%	74.2%	82.7%
	Piracicaba Dataset	Patches	71.5%	83.0%	93.0%
		Patient	94.3%	91.0%	96.0%
		RoI	91.0%	92.6%	97.9%
VGG-16	Sheffield Dataset	Patches	71.2%	72.4%	80.8%
		Patient	79.9%	90.4%	79.3%
		RoI	76.2%	70.7%	81.0%
	Piracicaba Dataset	Patches	83.1%	78.3%	95.7%
		Patient	97.9%	100.0%	95.4%
		RoI	94.6%	90.2%	100.0%

**Table 6 cancers-13-01291-t006:** Train and test with independent datasets results.

		Train	Validation	Test Patch	Test Patient
Inception ResNet-v2	Sheffield Trained Piracicaba Tested	81.9%	81.0%	71.7%	86.5%
	Piracicaba Trained Sheffield Tested	85.3%	84.1%	60.8%	66.7%
ResNet-101	Sheffield Trained Piracicaba Tested	99.4%	99.4%	70.0%	79.3%
	Piracicaba Trained Sheffield Tested	93.9%	93.4%	50.0%	75.8%

## Data Availability

The data that support the findings of this study are openly available in Zenodo at https://doi.org/10.5281/zenodo.4549721. Uploaded on 18 February 2021.

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
