# Peer review of "Convolutional Neural Network-Based Clinical Predictors of Oral Dysplasia: Class Activation Map Analysis of Deep Learning Results"

_cancers, 2021, doi:10.3390/cancers13061291_

Round 1

Reviewer 1 Report

Most of my concerns have been addressed. Remaining minor issues:

- Figure 4 is still confusing. The "Normal Region" is 464 x 510 pixels, so if the stride is 64 pixels, there should be 7 vertical and 7 horizontal dashed lines, but the figure shows 4 and 5, respectively. Similar problem for the "Lesion Region": the number of lines does not correspond to the reported patch size of 675 x 1394 pixels. The geometry of this illustration is wrong.

- Section 4: "However, for both of the results performed on these pre-trained networks, the system is more accurate when the Sheffield dataset was used for training and the Piracicaba dataset was used as testing." The authors' explanation for this, given in their response letter, should be added to the manuscript. Something along the lines of: "An explanation for this is that when the Sheffield dataset is used as training, the system is trained on relatively lower quality and more challenging images, and the resulting classifier works well on the higher-quality Piracicaba images. However, when the system is trained with better quality images, its performance is lower for the relatively lower quality images."

- Subfigures should be named consistently and referenced accordingly in the figure captions. Specifically, in Figure 6 you have removed "a" and "b" from the caption and now use "left panel" and "right panel", while in Figure 7 you have instead added "a", "b", "c" to the panels. Use either one of the two approaches (the journal's preferred style) consistently throughout the manuscript.

The writing can be improved in several places:

- Section 2.4: "retrained stated in the Table 2" > "retrained are stated in the Table 2".

- Figure 4 caption: "patches.." > "patches."

- Figure 7 caption: "Severe part divided into patches and the 4 patches from severe part classified as Suspicious. In bounding box of the lesion also classified but severe part couldn’t define the severe part of the lesion as abnormal. However, patches define the lesion part better." This still sounds very cryptic. Please rephrase by writing normal/formal sentences.

Author Response

We thank the editor and the reviewer for their thoughtful comments and constructive criticism.  We have thoroughly responded to all the comments, and their questions have brought further clarity to the revised manuscript.

Comments:

  1. Figure 4 is still confusing. The "Normal Region" is 464 x 510 pixels, so if the stride is 64 pixels, there should be 7 vertical and 7 horizontal dashed lines, but the figure shows 4 and 5, respectively. Similar problem for the "Lesion Region": the number of lines does not correspond to the reported patch size of 675 x 1394 pixels. The geometry of this illustration is wrong.

We have now clarified the images in Figure 4.

  1. Section 4: "However, for both of the results performed on these pre-trained networks, the system is more accurate when the Sheffield dataset was used for training and the Piracicaba dataset was used as testing." The authors' explanation for this, given in their response letter, should be added to the manuscript. Something along the lines of: "An explanation for this is that when the Sheffield dataset is used as training, the system is trained on relatively lower quality and more challenging images, and the resulting classifier works well on the higher-quality Piracicaba images. However, when the system is trained with better quality images, its performance is lower for the relatively lower quality images."

Thank you for your contribution, we have added the explanation that you suggested.

  1. Subfigures should be named consistently and referenced accordingly in the figure captions. Specifically, in Figure 6 you have removed "a" and "b" from the caption and now use "left panel" and "right panel", while in Figure 7 you have instead added "a", "b", "c" to the panels. Use either one of the two approaches (the journal's preferred style) consistently throughout the manuscript.

We have corrected the Figure 6 and 9 with their captions as below:

Figure 6. a. Heat maps true classified for Suspicious b. Heat maps true classified for Normal cases.

Figure 9. a. Shows the Inception ResNet-v2 pre-trained network results for two different ‘Suspicious’ cases, which are not well defined. b. Shows the ResNet-101 pre-trained network results for the same images of the upper row, and the ResNet-101 heat map is better than the upper row results. However, there are still some parts that are unexpected on the heat map.

  1. The writing can be improved in several places:

- Section 2.4: "retrained stated in the Table 2" > "retrained are stated in the Table 2".

Thank you for the correction, we have correct

- Figure 4 caption: "patches.." > "patches."

Thank you for the correction, we have correct

- Figure 7 caption: "Severe part divided into patches and the 4 patches from severe part classified as Suspicious. In bounding box of the lesion also classified but severe part couldn’t define the severe part of the lesion as abnormal. However, patches define the lesion part better." This still sounds very cryptic. Please rephrase by writing normal/formal sentences.

We have clarified the caption of Figure 7 as:

“Original image was classified as ‘Suspicious’ when the region of interest was tested. However, the heat map of the region of interest did not reflect the severe part of the image with a warm color (e.g., red). On the other hand, when we divide the severe part of the image into four patches, the heat map of the patches indicates the severe region with an intense color between yellow and red.”

This manuscript is a resubmission of an earlier submission. The following is a list of the peer review reports and author responses from that submission.

Round 1

Reviewer 1 Report

The work presented by Camalan, et al on CNN methods for oral dysplasia presents a clear roadmap for the use of CNN as a screening tool. In particular, the strength of this manuscript include, 1) excellent graphical summary of how  CNN approach is processing the data; 2) discussion of misleading heat maps, which highlights the limitation of these approaches and are not always included research on CNN. The major limitation is as the authors concluded, the small number to train and test.

Major Comments

No major comments – the methodology is sound, great breakdown on the comparisons between ML approaches.

Minor Comments

  • Were other cut-offs considered for training vs validation cohorts? Why was the 85:15 approached utilized?
  • How do the authors envision the implementation as a clinical support tool?
  • How do the authors predict the inclusion of clinical variables will affect the performance?

Author Response

We thank the editor and the reviewers for their thoughtful comments and constructive criticism.  We have thoroughly responded to all the comments, and their questions have brought further clarity to the revised manuscript.

Reviewer's Responses to Questions

Reviewer 1:

The work presented by Camalan, et al on CNN methods for oral dysplasia presents a clear roadmap for the use of CNN as a screening tool. In particular, the strength of this manuscript include, 1) excellent graphical summary of how  CNN approach is processing the data; 2) discussion of misleading heat maps, which highlights the limitation of these approaches and are not always included research on CNN. The major limitation is as the authors concluded, the small number to train and test.

Major Comments

No major comments – the methodology is sound, great breakdown on the comparisons between ML approaches.

            Thank you for carefully reviewing the manuscript and for your encouraging and valuable comments.

Minor Comments

  • Were other cut-offs considered for training vs validation cohorts? Why was the 85:15 approached utilized?

We set aside 10% of the cases for independent testing. The remaining 90% was used for training and validation. Because we wanted to have at least 10% of the cases for validation and more cases in the validation set than the test set, 90% of the cases were divided into 85% for training and 15% for testing. This is all done while also trying to maximize the number of cases in the training set.

  • How do the authors envision the implementation as a clinical support tool?

This is a feasibility study with two independent datasets of clinical photographic images from two different countries (the UK and Brazil). When fully developed and validated, the system could be used as part of clinical exams. Keeping this in mind, we designed the system to use regular photographic images instead of sophisticated and expensive imaging modalities. Because early detection is critical to improving survival, patient with images that are classified as ‘suspicious’ can be followed up or urgently referred for further specialist input and/or a biopsy.

  • How do the authors predict the inclusion of clinical variables will affect the performance?

We expect that including patients’ age, gender, medical history, smoking, and alcohol consumption habits is likely to increase the system's classification accuracy. However, including these variables would require a much larger database and ‘fusion of the clinical data with the images’ is a key elements of this study's future work.

Reviewer 2 Report

This paper addresses the problem of early oral cancer detection and explores the potential of various deep neural networks applied to photographic oral images for this purpose. A CNN-based transfer learning approach is implemented using the Inception-ResNet-v2 pretrained network which is compared to VGG-16, Resnet-101, and Inception-v3. The presented experimental results on two datasets from different institutions are promising. Methodologically there is not much new. All networks are known and their use is rather straightforward. The main technical innovation is the use of heat maps to analyze which regions in the images contribute most to the predictions. Such maps could help in understanding the decision making process of the networks. Overall this is an interesting study that seems to be the first of its kind for the particular application. It does, however, suffer from many shortcomings, as listed below.

- Section 2.1: "The images were standard colour images captured with a regular photographic camera..." There can be huge differences between cameras, and image quality may vary widely even with the same camera from one shot to the next depending on various external factors, so details matter here. To facilitate reproducibility, please provide details about the camera brand, type, resolution (the image sizes mentioned in Table 1 are not standard, so how did you get to these?), other relevant image quality settings, and whether they were used consistently throughout the project and between institutions or varied. If different cameras were used, that is also important to know. Some information about the variability of the data is helpful in appreciating the robustness of the proposed method.

- Table 1: "Number of Case" > "Number of Cases" (2x).

- Figure 4: The numbers and lines in the figure are confusing. For example, the "Normal Region" is said to be of size "464 x 510" (I assume this means pixels), which is not a multiple of "128 x 128" (the patch size), but the black lines raster does suggest this. Same comment for the "Lesion Region". Please clarify.

- Section 2.2: "data augmentation was performed..." and then the types of transformations applied are listed. Do I understand correctly that these were always applied individually? Did you also try combining multiple transformations? For example, a random shift and then a random scaling afterwards? That would give even more data.

- Figure 5: Rotation and sharpening are mentioned as augmentation approaches but these are missing in the text. Please clarify.

- Section 2.2: "random horizontal and vertical flips". Why/how random? Do you mean for each patch it was randomly decided whether to flip horizontally or vertically? Why not always both (not random)? That would yield more data.

- Section 2.2: "elastic transformation (alpha ranges between 45 to 100 and sigma is 5)". It is not clear what kind of transformation this is, exactly, or what the parameters alpha and sigma mean here. Please clarify and ideally add a literature reference for details.

- Section 2.4: "Welikala et al..." Add the reference number [23].

- Section 2.4: "Instead, we opted to freeze the first 818 layers, a number that was decided empirically to 257 limit the number of parameters required to learn the features in the network." It would be insightful to provide the actual numbers of parameters for all networks. Please add a table giving for each network the total number of parameters and the actual number of parameters retrained by you. This gives an impression of the total learning capacity of the different types of networks versus the partial capacity that was actually used to further train the networks on your data.

- Setion 3: "The ten-fold minimum test results for the Piracicaba dataset are higher than those of the Sheffield dataset and closer to the validation and train accuracies. These results mean that the classifiers trained on the Sheffield dataset are prone to overfitting." More generally, it seems the networks perform quite differently on the two datasets. More discussion is needed explaining what may be the cause of this.

It is stated that "This difference in accuracy may be explained by the fact that the Sheffield dataset has a smaller number of images and ROIs." This hypothesis can be tested fairly easily by reducing the number of patches in the Piracicaba dataset to that of the Sheffield dataset and seeing whether this indeed reduces the performance of the networks on the former to the level of the latter.

It is also stated that "The image dimensions are smaller, affecting the sizes of suspicious lesions." This, too, can be tested by downscaling the dimensions of the Piracicaba dataset to those of the Sheffield dataset and repeating the experiments to see whether this indeed reduces the performance accordingly.

- Section 3: "These results mean that the classifiers trained on the Sheffield dataset are prone to overfitting." That sounds plausible, but then the results of Table 5 are confusing: the networks trained on the Sheffield dataset give higher performance on the Piracicaba dataset than vice versa. Some explanation is needed.

- Page 10: "visa-versa" > "vice versa".

- Table 4: The results for Inception ResNet-v2 and ResNet-101 on the Piracicaba dataset at the Patient level are exactly the same (94.0%, 98.0%, 93.0%). Is that a coincidence? These networks are not quite the same, so this needs an explanation.

- Figure 6 caption: "a. Suspicious and b. Normal Cases". There is no "a" and "b". Better write "Suspicious (left panel)" and "Normal (right panel)".

- Page 11: "While the heat map correctly identifies all the correct areas in Figure 6 for both normal and suspicious cases." This sentence seems truncated. Please fix.

- Figure 7 caption: "a", "b", "c" are not in the figure. Please fix.

Also, remove the text from the figure itself and put it in the caption. The text is rather cryptic. Please improve this description of what is going on.

Another problem is that two of the four patches are overlapping. Is that supposed to be the case? It is not clear from the method description that you sample overlapping patches. Figure 4 suggests they are adjacent and not overlapping.

- Figure 8: There is no "a" and "b" in the figure.

- Section 4: "RGB images". This is the first time the term is used in the paper. It should be mentioned already in the methods section.

- Section 4: "The F1-score ranges from 71.1% to 97.9%". Not clear what you mean. In Table 4 the minimum F1-score is 70.0%.

Also, it is stated "recall and precision varies between 67.3% to 100%, and 78.2% to 100%, respectively", but in Table 4 the recall starts from 67.0% and the precision from 78.0%.

And "The highest F1-score, precision, and recall values are obtained for the Piracicaba dataset with VGG-16 and ResNet-101 pre-trained models for the patient-level tests..." is not exactly true. For example, Inception-v3 has a higher precision at the patient level for the Piracicaba dataset than both VGG-16 and ResNet-101. Since the F1-score is the harmonic mean of the recall and precision and thus summarizes the two, it is better to just focus on the F1-score in the discussion. And then you have to conclude that VGG-16 gives the best overall performance on that dataset, and the other three networks perform equally well. At the ROI level, ResNet-101 performs best in terms of F1-score.

These observations make the discussion section rather questionable. All the findings should be carefully reconsidered and the discussion rewritten accordingly.

- Section 4: "the results are promising, especially considering that they were developed and tested". It is a bit strange to say that results were developed and tested. This should read "the results are promising, especially considering that the networks were developed and tested".

- References [1][2][3]: Fix problems with author names and journal titles.

Author Response

We thank the editor and the reviewers for their thoughtful comments and constructive criticism.  We have thoroughly responded to all the comments, and their questions have brought further clarity to the revised manuscript.

Reviewer 2:

This paper addresses the problem of early oral cancer detection and explores the potential of various deep neural networks applied to photographic oral images for this purpose. A CNN-based transfer learning approach is implemented using the Inception-ResNet-v2 pretrained network which is compared to VGG-16, Resnet-101, and Inception-v3. The presented experimental results on two datasets from different institutions are promising. Methodologically there is not much new. All networks are known and their use is rather straightforward. The main technical innovation is the use of heat maps to analyze which regions in the images contribute most to the predictions. Such maps could help in understanding the decision making process of the networks. Overall this is an interesting study that seems to be the first of its kind for the particular application. It does, however, suffer from many shortcomings, as listed below.

- Section 2.1: "The images were standard colour images captured with a regular photographic camera..." There can be huge differences between cameras, and image quality may vary widely even with the same camera from one shot to the next depending on various external factors, so details matter here. To facilitate reproducibility, please provide details about the camera brand, type, resolution (the image sizes mentioned in Table 1 are not standard, so how did you get to these?), other relevant image quality settings, and whether they were used consistently throughout the project and between institutions or varied. If different cameras were used, that is also important to know. Some information about the variability of the data is helpful in appreciating the robustness of the proposed method.

            We have now clarified the sentences as:

The images were standard colour images captured with regular photographic cameras, which are defined and detailed in the appendix, capturing the lesion as well as the surrounding regions of the mouth.

Appendix A

Sheffield Camera Details:

The standard dental views:

  • AP or frontal view with retractors in place and patient biting together.     Lens magnification 1:2  
  • 2 buccal views with retractors and patient biting down                             Lens magnification 1:2
  • Upper and lower biting surfaces                                                             Lens magnification 1:2.5

The lens magnifications apply to film and full frame digital cameras

The equipment used:

  • Cameras          
    • Nikon D200 and Nikon D800 (Sheffield)
    • Nikon D100 and Canon EOS 7D (Piracicaba)
  • Lens                
    • Nikon 105mm and Sigma 105mm (Sheffield)
    • Nikon 105mm and Canon EF 100mm (Piracicaba)
  • Ring Flash       
    • Nikon and Sigma (Sheffield)
    • Canon (Piracicaba)

Nikon D200 camera used APS-c sized sensors, for this reason the standard distance settings were adjusted to ensure the images magnification was maintained.        

Nikon D800 camera were full framed digital SLR and allowed us to revert back to standard magnification setting that were used when film cameras were used. 

Image Quality

            There would be slight variations but as most of the settings are standardised we do not envisage any major image quality issues. The photography departments have working protocols in place and a policy of not making any significant changes to images, which would enable as much consistency as possible to be achieved.

            The only adjustment needed for digitally captured images in some instances is the lighting/brightness adjustment.  To maintain image size there is no cropping.

Training

            The medical illustration departments adopt the guidance from the Institute for Medical Illustrators (IMI) which also prescribes standard views and magnifications. All photographers are trained to adopt the IMI method of working.

- Table 1: "Number of Case" > "Number of Cases" (2x).

            Thank you, it is corrected.

- Figure 4: The numbers and lines in the figure are confusing. For example, the "Normal Region" is said to be of size "464 x 510" (I assume this means pixels), which is not a multiple of "128 x 128" (the patch size), but the black lines raster does suggest this. Same comment for the "Lesion Region". Please clarify.

            We have now clarified the image and the caption as:

Figure 4. Figurative image patching from bounding box lesion and normal area into 128x128 patches. Original images were annotated by dentists. We cropped the bounding box of the annotated regions as normal or suspicious regions. The cropped regions were divided into 128x128 pixel image patches (red and blue squares) with a stride of 64 pixels (dotted lines), resulting in overlapping patches. If the size of the image was not a multiple of 128x128, the patches were not obtained from the remainder of the image. If the patches with more than 80% pixels originated from a suspicious area, they were labeled as suspicious patches; otherwise, they were considered as normal patches.

You can also see the image in the word document.

- Section 2.2: "data augmentation was performed..." and then the types of transformations applied are listed. Do I understand correctly that these were always applied individually? Did you also try combining multiple transformations? For example, a random shift and then a random scaling afterwards? That would give even more data.

            That is correct; we applied each augmentation individually. We did not combine multiple transformations, but it would be a good idea to apply combinations. Thank you for your suggestion, and we have added this suggestion to our future work.

- Figure 5: Rotation and sharpening are mentioned as augmentation approaches but these are missing in the text. Please clarify.

            We thank the reviewer for pointing this out. We have now clarified the sentences:

Both for suspicious and normal patches, data augmentation was performed with random horizontal and vertical flips, randomly shifting an input patch in the horizontal and vertical directions with two offsets sampled uniformly between −50 and 50 pixels, rotating the images starting from -5 to 5 and increase by 5 up to -45 to 45 angles, sharpening images (alpha ranges between 0 to 1 and lightness is 0.75 and 1.5,  randomly scaling the inputs by a factor between 0.5 and 1.5, and elastic transformation (alpha ranges between 45 to 100 and sigma is 5) [24].

- Section 2.2: "random horizontal and vertical flips". Why/how random? Do you mean for each patch it was randomly decided whether to flip horizontally or vertically? Why not always both (not random)? That would yield more data.

            We used both horizontal and vertical flips and randomly decided to increase the variability.

- Section 2.2: "elastic transformation (alpha ranges between 45 to 100 and sigma is 5)". It is not clear what kind of transformation this is, exactly, or what the parameters alpha and sigma mean here. Please clarify and ideally add a literature reference for details.

            We have clarified the sentence as mentioned above, and we added the citation:

    ‘P. Y. Simard, D. Steinkraus, and J. C. Platt, "Best practices for convolutional neural networks applied to visual document analysis," in Icdar, 2003, vol. 3, no. 2003: Citeseer.’

This citation defines the transformation. We also added to the text that we used the predefined ‘imgaug’ library for image augmentation in Python.

- Section 2.4: "Welikala et al..." Add the reference number [23].

            This reference has been added.

- Section 2.4: "Instead, we opted to freeze the first 818 layers, a number that was decided empirically to 257 limit the number of parameters required to learn the features in the network." It would be insightful to provide the actual numbers of parameters for all networks. Please add a table giving for each network the total number of parameters and the actual number of parameters retrained by you. This gives an impression of the total learning capacity of the different types of networks versus the partial capacity that was actually used to further train the networks on your data.

We have clarified the ‘The number of frozen layers and the number of parameters that are retrained stated in Table 2’ caption and added the Table 2.

Table 2. Number of retrained Cross-Validation Results for Sheffield and Piracicaba Datasets.

Pre-trained

Network

Total Number

The Number of

Layers

Parameters (Millions)

Frozen Layers

Retrained Parameters (Millions)

Image Size

Inception ResNetv2

825

55.9

818

1.5

299x299x3

Inception-v3

315

23.9

307

2.05

299x299x3

ResNet-101

347

44.6

79

43.2

224x224x3

VGG-16

41

138

0

138

224x224x3

- Setion 3: "The ten-fold minimum test results for the Piracicaba dataset are higher than those of the Sheffield dataset and closer to the validation and train accuracies. These results mean that the classifiers trained on the Sheffield dataset are prone to overfitting." More generally, it seems the networks perform quite differently on the two datasets. More discussion is needed explaining what may be the cause of this.

Thank you for this question. There is a typo in ‘minimum test results;’ it should say ‘average test results,’ which is more meaningful. This has been corrected. ‘Minimum’ values can change according to randomly selected patients. Therefore, ‘average’ is a reliable measurement for accuracy.

It is stated that "This difference in accuracy may be explained by the fact that the Sheffield dataset has a smaller number of images and ROIs." This hypothesis can be tested fairly easily by reducing the number of patches in the Piracicaba dataset to that of the Sheffield dataset and seeing whether this indeed reduces the performance of the networks on the former to the level of the latter.

We agree with the reviewer that there are many factors that may explain the performance numbers; the difference in the number of images and ROIs is just one of them. The other factors are image size, quality, and where these patches are selected. In order to be more comprehensive, we have changed this sentence as follows:

This difference in accuracy may be explained by many factors, including the differences in the number of images, ROIs, image size and quality, and where the patches are selected.

It is also stated that "The image dimensions are smaller, affecting the sizes of suspicious lesions." This, too, can be tested by downscaling the dimensions of the Piracicaba dataset to those of the Sheffield dataset and repeating the experiments to see whether this indeed reduces the performance accordingly.

We have already scaled the images in the range of 0.5 to 1.5 during the augmentation process for both datasets.

However, it may not be appropriate to conduct controlled experiments with relatively small size datasets and to derive general conclusions about the effect of image sizes or other factors. 

- Section 3: "These results mean that the classifiers trained on the Sheffield dataset are prone to overfitting." That sounds plausible, but then the results of Table 5 are confusing: the networks trained on the Sheffield dataset give higher performance on the Piracicaba dataset than vice versa. Some explanation is needed.

We are sorry for the confusion. When the Sheffield dataset is used as training, the system is trained on relatively lower quality and more challenging images, and the resulting classifier works well on Piracicaba images. However, when the system is trained with better quality images, its performance is lower for the relatively lower quality images.  

- Page 10: "visa-versa" > "vice versa".

            This has been corrected.

- Table 4: The results for Inception ResNet-v2 and ResNet-101 on the Piracicaba dataset at the Patient level are exactly the same (94.0%, 98.0%, 93.0%). Is that a coincidence? These networks are not quite the same, so this needs an explanation.

Thank you for this question. There are some typos that were a result of the table being copied, and the decimal place was rounded to the nearest number. It has now been corrected in the revised manuscript. Whilst they are not exactly the same, the values are still close.

Table 4. Leave one patient out Cross-Validation F1-score, Recall, and Precision for Sheffield and Piracicaba Datasets.

F1-score

Recall (Sensitivity)

Precision (PPV)

Inception ResNet-v2

Sheffield Dataset

Patches

77.9%

74.1%

90.2%

Patient

87.2%

99.3%

81.1%

RoI

87.0%

77.8%

97.6%

Piracicaba  Dataset

Patches

69.8%

66.9%

85.8%

Patient

94.2%

97.9%

92.5%

RoI

89.4%

82.1%

98.2%

ResNet-101

Sheffield Dataset

Patches

71.1%

72.0%

79.0%

Patient

84.0%

98.0%

78.2%

RoI

80.2%

73.0%

87.0%

Piracicaba  Dataset

Patches

85.0%

67.3%

89.1%

Patient

94.5%

98.5%

94.9%

RoI

96.3%

84.8%

98.4%

Inception-v3

Sheffield Dataset

Patches

69.9%

71.3%

78.4%

Patient

85.1%

95.8%

80.8%

RoI

78.3%

74.2%

82.7%

Piracicaba  Dataset

Patches

71.5%

83.0%

93.0%

Patient

94.3%

91.0%

96.0%

RoI

91.0%

92.6%

97.9%

VGG-16

Sheffield Dataset

Patches

71.2%

72.4%

80.8%

Patient

79.9%

90.4%

79.3%

RoI

76.2%

70.7%

81.0%

Piracicaba  Dataset

Patches

83.1%

78.3%

95.7%

Patient

97.9%

100.0%

95.4%

RoI

94.6%

90.2%

100.0%

- Figure 6 caption: "a. Suspicious and b. Normal Cases". There is no "a" and "b". Better write "Suspicious (left panel)" and "Normal (right panel)".

            This has been corrected.

- Page 11: "While the heat map correctly identifies all the correct areas in Figure 6 for both normal and suspicious cases." This sentence seems truncated. Please fix.

            The sentences have been corrected as follows:

While the heat map correctly identifies all the correct areas as in Figure 6 for both normal and suspicious cases, in some samples, the results of the heat map are somewhat misleading, for example, in Figure 7.

- Figure 7 caption: "a", "b", "c" are not in the figure. Please fix.

Also, remove the text from the figure itself and put it in the caption. The text is rather cryptic. Please improve this description of what is going on.

Figure 7 has been corrected, and ‘a, b, c’ captions are now included with the images. We have removed the text from the figure and added it to the caption.

Another problem is that two of the four patches are overlapping. Is that supposed to be the case? It is not clear from the method description that you sample overlapping patches. Figure 4 suggests they are adjacent and not overlapping.

            We have fixed Figure 4.  The patches are overlapping; we have now clarified this in the text.

- Figure 8: There is no "a" and "b" in the figure.

Caption ‘a’ and ‘b’ are added to Figure 8.

- Section 4: "RGB images". This is the first time the term is used in the paper. It should be mentioned already in the methods section.

‘RGB’ has removed from the text

- Section 4: "The F1-score ranges from 71.1% to 97.9%". Not clear what you mean. In Table 4 the minimum F1-score is 70.0%.

Also, it is stated "recall and precision varies between 67.3% to 100%, and 78.2% to 100%, respectively", but in Table 4 the recall starts from 67.0% and the precision from 78.0%.

            All tables have been fixed, and the caption of Table 5 now reads as follows:

The F1-score ranges from 69.8% to 97.9%; recall and precision varies between 67.3% to 100%, and 78.2% to 100%, respectively.

And "The highest F1-score, precision, and recall values are obtained for the Piracicaba dataset with VGG-16 and ResNet-101 pre-trained models for the patient-level tests..." is not exactly true. For example, Inception-v3 has a higher precision at the patient level for the Piracicaba dataset than both VGG-16 and ResNet-101. Since the F1-score is the harmonic mean of the recall and precision and thus summarizes the two, it is better to just focus on the F1-score in the discussion. And then you have to conclude that VGG-16 gives the best overall performance on that dataset, and the other three networks perform equally well. At the ROI level, ResNet-101 performs best in terms of F1-score.

These observations make the discussion section rather questionable. All the findings should be carefully reconsidered and the discussion rewritten accordingly.

            Thank you for your comment and recommendations. We have clarified the section as:

            The highest F1-score is obtained for the Piracicaba dataset with VGG-16 pre-trained models for overall performance, and the other three pre-trained networks performed similarly well. For the RoI-level results, the best F1-score was obtained with ResNet-101.

- Section 4: "the results are promising, especially considering that they were developed and tested". It is a bit strange to say that results were developed and tested. This should read "the results are promising, especially considering that the networks were developed and tested".

            This has been corrected.

- References [1][2][3]: Fix problems with author names and journal titles.

This has now been corrected.
